# Identification of *VvAGL* Genes Reveals Their Network’s Involvement in the Modulation of Seed Abortion via Responding Multi-Hormone Signals in Grapevines

**DOI:** 10.3390/ijms25189849

**Published:** 2024-09-12

**Authors:** Fei Liu, Rana Badar Aziz, Yumiao Wang, Xuxian Xuan, Mucheng Yu, Ziyang Qi, Xinpeng Chen, Qiqi Wu, Ziyang Qu, Tianyu Dong, Shaonan Li, Jinggui Fang, Chen Wang

**Affiliations:** College of Horticulture, Nanjing Agricultural University, Nanjing 210095, China; 2021104030@stu.njau.edu.cn (F.L.); azizranabadar@stu.njau.edu.cn (R.B.A.); 2021104029@stu.njau.edu.cn (Y.W.); 2021204006@stu.njau.edu.cn (X.X.); 2022804156@stu.njau.edu.cn (M.Y.); 2022804173@stu.njau.edu.cn (Z.Q.); xinpengchen2000@163.com (X.C.); wuqqi883246@163.com (Q.W.); qzyang@njau.edu.cn (Z.Q.); dtyu0828@gmail.com (T.D.); lshnan666@gmail.com (S.L.); fangjj1215@163.com (J.F.)

**Keywords:** grape seed abortion, *VvAGL* sub-family, multi-hormone pathway, regulatory network

## Abstract

The formation of seedless traits is regulated by multiple factors. AGLs, which belong to the MADS-box family, were reported to be important regulators in this process; however, the underlying mechanism remains elusive. Here, we identified the VvAGL sub-family genes during the seed abortion process in seedless grapevine cv. ‘JingkeJing’ and found 40 differentially expressed VvAGL members and 1069 interacting proteins in this process. Interestingly, almost all members and their interacting proteins involved in the tryptophan metabolic pathway (K14486) and participated in the phytohormone signalling (KO04075) pathway, including the growth hormone (IAA), salicylic acid (SA), abscisic acid (ABA), cytokinin (CTK), and ethylene signalling pathways. The promoters of AGL sub-family genes contain cis-elements in response to hormones such as IAA, ABA, CTK, SA, and ETH, implying that they might respond to multi-hormone signals and involve in hormone signal transductions. Further expression analysis revealed *VvAGL6-2*, *VvAGL11, VvAGL62-11*, and *VvAGL15* had the highest expression at the critical period of seed abortion, and there were positive correlations between ETH-*VvAGL15*-*VvAGL6-2*, ABA-*VvAGL80*, and SA-*VvAGL62* in promoting seed abortion but negative feedback between IAA-*VvAGL15*-*VvAGL6-2* and CTK-*VvAGL11*. Furthermore, many genes in the IAA, ABA, SA, CTK, and ETH pathways had a special expressional pattern in the seed, whereby we developed a regulatory network mediated by VvAGLs by responding to multihormonal crosstalk during grape seed abortion. Our findings provide new insights into the regulatory network of VvAGLs in multi-hormone signalling to regulate grape seed abortion, which could be helpful in the molecular breeding of high-quality seedless grapes.

## 1. Introduction

Grapevine (*Vitis vinifera* L.) is one of the most important economic fruit trees cultivated widely in the world [1]. Since seedless berries are convenient to eat and process and as well possess a good flavour quality, they are widely used in the fresh food, drying, and processing industry, and, thus, are very popular with people [2]. Driven by market demand, seedless breeding has become one of the significant objectives of grape breeding. However, the normal cross-breeding process of grapes requires too much time and more labour, and the efficiency rate of its breeding is very low. Moreover, the seedless grape cultivars available cannot meet market demand. With the development of molecular biotechnology, molecular designing breeding and genetic engineering breeding have been developed as novel breeding strategies for solving the above problem in the grape industry, while these strategies rely entirely on gaining insight into the molecular mechanism of the formation of the grape seedless traits [3].

Based on the conditions of pollination and fertilisation, seedless grapes can be classified into three different parthenocarpy types: stimulated parthenocarpy, pseudo-parthenocarpy, and parthenocarpy [4]. During the parthenocarpy process, the berry develops from maternal tissues without the pollination and fertilisation of egg cells, and this type of berry is difficult to use for breeding; however, parthenocarpy can be induced using artificial measures. Moreover, the development of seedless fruits without fertilisation in the ovary under the treatment of external environmental conditions (mainly low temperature) or exogenous growth regulators, which is called stimulated-parthenocarpy, and this type is utilised widely in the production of seedless grapes, while pseudo-parthenocarpy is a phenomenon in which the ovary can be normally fertilised but is affected by its own or external conditions, resulting in embryo abortion to produce seed scars or smaller seeds, while parts such as the ovary or receptacle can continue to develop and, ultimately, form seedless fruits [5]. Pseudo-parthenocarpy traits can be inherited by the next generation, and pseudo-parthenocarpic seed abortion is the main type of seedless grapes, accounting for about 85% [6]. Moreover, the ease of consumption and the convenience of drying processes have made seedless grapes popular among the general public. More than 98% of seedless grape types are employed in dry processing, and more than 80% of all table grapes are seedless. Therefore, the selection and breeding of high-quality new varieties of seedless grapes has become an attractive direction for the development of the grape industry as well as breeding research [7].

The MADS-box transcription factors widely exist in higher plants with a large and diverse family [8,9]. They are the transcription factors that have extremely important roles in the development of plants, which include not only the inflorescence and flowers but also fruit and pollen [10]. Meanwhile, MADS-box genes participate in developmental changes accompanying fruit maturity and ripening [11]. A previous study covering the MADS-box family in grapevine demonstrated that *VvAGL11* is a strong candidate gene involved in the absence of seeds [12,13] due to the missense mutation of Arg at position 197 to Leu [14]. More recently, another study showed that *VvAGL11* is 25 times more expressed in fruits at the young fruit stage compared with flower stages; however, the expression is decreased in seed tissues in seedless grapevines [15]. Furthermore, it is repressed in roots, branches, leaves, buds, and tendrils [16]. The *VvAGL11* is a member of the MADS-box transcription factor (TF) family reported for a broad range of species [17]. *VvAGL11* has been described as a homolog to SEEDSTICK (STK) or AGAMOUS-LIKE 11 (AGL11), which specifically acts on the identity and control of ovule and seed development in *Arabidopsis thaliana* [18,19]. The *AtAGL11* mutant SEEDSTICK (STK) presents a reduced number and size of seeds [20,21]. Natural loss-of-function alleles of *VvAGL11* have been implicated as the major determinant of seed abortion. Several studies have demonstrated that the down-regulation of *SlAGL11* in tomatoes results in seed shrinkage and the overexpression of *SlAGL11* transformed sepals into fleshy organs [22]. Meanwhile, the expression level of *AGL11* was restricted in naturally seedless grape seeds. Moreover, the overexpression of oil seed rape *BnAGL11* resulted in smaller curled leaves and accelerated leaf senescence [23]. *ZaAGL11* may reduce the number and weight of fruits and seeds [24]. The expression of *Arabidopsis* MADS-box transcription factor *AtAGL62* also leads to reduced seed size [25]. It was found that *AGL91* and *AGL40* influence seed size by regulating endosperm development [26]. *AGL67* inhibits seed germination by activating *SOM* expression [27]. *CnMADS1* is an AGAMOUS (AGL)-like MADS-box transcription factor in coconut endosperm, and heterologous overexpression of this gene resulted in a significant increase in both seed weight and volume [28]. Indeed, MADS-box genes also play an important role in regulating the development of floral organs [29,30]. *VvMADS45* regulates the development of floral organs and ovules [31]. All these studies confirm that the *AGL* sub-family could participate in multi-biological processes, especially in seed development. Most of the studies on seed abortion are related to gene function and phenotypic aspects, and the differences in transcriptional expression levels are still unclear; therefore, our study was developed based on this objective.

Recently, in this work, we found that the *VvAGL* sub-family, especially *VvAGL6*-2, *VvAGL11*, and *VvAGL62*, might play important roles in seed abortion and be involved in the regulation of seed abortion by responding to hormonal signals; however, how this sub-family responds to hormonal signals to regulate grape seed abortion and their regulatory network are unclear. Therefore, we carried out the characterisation of the promoters of *AGL* sub-family genes, prediction of interacting proteins, analysis of hormone response and expression pattern, and illuminated the *AGL* sub-family-mediated regulatory network in grape seeds via interacting proteins through guiding multi-hormone signalling pathways and identified the critical modulators in this process. Our findings could be helpful in enriching the knowledge of modulation of grape pseudo-parthenocarpy and provide important support for gaining further insight into the molecular mechanism of seedless trait formation in grapes and have implications for the breeding of seedless grapes with high quality.

## 2. Results

### 2.1. Developmental Changes in Berries during the Abortion of Grape Seeds

We assessed the dynamics of berries and seeds during berry development in ‘Zhengyan seedless’ grapes and revealed the severe inhibition of its seeds’ development, resulting in the forming of seedless berries. Regarding seed sizes, there were almost no changes during the whole berry development (Figure 1A), but the vertical and horizontal diameters of ‘Zhengyan seedless’ grape berries increased significantly across developmental periods (Figure 1B).

### 2.2. Identification and Characterisation of VvAGL Sub-Family Genes in Grape Seeds

Based on their sequential placements on the chromosome, the 40 *AGL* family genes that were found in the seed abortion grape genome are referred to as *VvAGL1~Vv AGL92*. Respectively, the amino acid sequence comparison analysis revealed that the *VvAGL* protein sequence similarity was 29.77% and that all the 40 *VvAGLs* contained the MADS structural domains that characterise the MADS-box gene family. Appendix A shows that grape *AGL* sub-family gene members were distributed on chromosomes 0 to 18—chromosome 3 had the most gene members. The theoretical isoelectric points of *AGL* sub-family gene members ranged from 4.62 (*VvAGL17*) to 5.40 (*VvAGL6-1*), all 40 *AGL* members encoded acidic amino acids, the protein fat coefficients ranged from 20.65 to 36.35, and the overall thermal stability was low; the hydrophilicity ranged from −0.009 to 0.847, which indicated that most of *AGL* proteins are hydrophobic proteins (Figure 2); the results of the physicochemical property analyses indicated that the differences in properties among the different genes of the AGL sub-family suggest differences in physiological functions among the genes.

To identify the regulatory role of the *VvAGL* sub-family during grape seed abortion, we first identified and characterised all members of the AGL sub-family in grape-aborted seeds. A total of 40 *VvAGL* members were identified using RNA-seq data from grape seed abortion tissues. Their encoded proteins were further analysed using MEGA 11.0 software. Based on evolutionary tree branching and gene structure analysis of *VvAGLs*, we categorised these genes into three groups, with group I having the most members (Figure 3A). In addition, we analysed the sequence structures of *VvAGLs* using GSDS online software (http://gsds.cbi.pku.edu.cn, accessed on 4 August 2023) (Figure 3B) and generally observed some similarity in the sequence structures belonging to the same group; moreover, *VvAGL* members had short and few introns, and *VvAGL6-1* had the fewest and shortest exon with only one, while group II showed that *VvAGL* members had longer introns and a relatively loosely packed structure. *VvAGL21* members had the longest sequence among all genes; group III showed that *VvAGL* members had shorter introns and a relatively cohesive structure.

In addition, map-checking software (http://mg2c.iask.in/mg2c_v2.0/, accessed on 5 August 2023) was used to localise 40 *VvAGL* genes on chromosomes. *VvAGL* was distributed on 13 of the 19 grapevine chromosomes (Figure 3C). *VvAGL* members were unequally located on each chromosome, with six *VvAGL* members on chr 0, chr15, seven *VvAGL* members on chr3, four on chr10, four on chr10, four on chr2, three on chr2 and chr5, two *VvAGL* members on chr1, 7, 8, and 14, and only one member on the remaining chromosomes, implying that the distribution of *AGL* sub-family is uneven in chromosomes, of which some members possess the positional conservatism.

### 2.3. Conservative and Evolutionary Analysis of the VvAGL Family across Diverse Plant Species

Using grape, *Arabidopsis thaliana*, apple, tomato, citrus, and corn *AGL* proteins to construct a phylogenetic tree, it was found that 177 proteins from five species could be clustered into five sub-families with various numbers of members, of which group II had the most members with 12, followed by group I (11), group III (7), and group IV (6), while group V had the least amount of members, with only three members (Figure 4). The same group possess similar sequence structures, indicating that their members might be conservative in structures and might possess similar potential functions. Moreover, the *VvAGL* family members are closely related to the branches of apple and tomato, indicating that they have a closer relationship. These results suggest that different genes in the same group have similar gene structures and are closely related to each other.

### 2.4. Gene Ontology and Pathway Mapping of VvAGLs

According to the analysis of the GO and KEGG pathways of the VvAGL family in grape seed abortion, we found that 40 members of the VvAGL family are involved in phytohormone signalling (ko04075) and the tryptophan metabolism pathway (K14486), affecting phytohormone biosynthesis (Table 1). The majority of the members of the entire gene sub-family were found to be involved in the tryptophan metabolic pathway, indicating that they have potential functions that are largely similar to those related to amino acid synthesis and phytohormone metabolism. These findings were also supported by GO analysis; 11 members of VvAGLs were found to be involved in the hormone signalling pathway of gene expression regulation (GO: 0010468), together with KEGG pathway analysis, deducing that VvAGLs might regulate grape seed abortion in response to multi-hormone crosstalk. In addition, the expression of the 40 VvAGL members varied during grape seed abortion, implying that members of the VvAGL sub-family differ in their ability to regulate grape seed abortion.

To more comprehensively recognise the potential functions of the VvAGLs sub-family, we further investigated the molecular interactions, reactions, and regulatory networks of their interacting proteins. The results showed that a total of 1069 interacting proteins were identified in STRING v10.0 at different developmental periods, among which the most interacting proteins were 395 in the JS vs. JB period, 377 in the JY vs. JS period, and the least interacting proteins (297) were identified in the JY vs. JB period (Figure 5C). Through KEGG analysis, we found that these interacting proteins were involved in 24 KEGG pathways. The pathway with the most interacting proteins was phytohormone signalling (ko04626), with 23 members (Figure 5B). Moreover, these interacting proteins were mainly derived from *VvAGL6*, *VvAGL15*, *VvAGL62*, and *VvAGL80*, suggesting that they may be the major regulators of the hormone signalling pathway in grape seed development. Meanwhile, using GO analysis, we found that VvAGLs might be involved in 14 biological processes, of which 14 genes were involved in the hormone-mediated signalling pathway (GO: 0010468) cellular macromolecular biosynthesis process (GO: 0034645) (Figure 5A), and these interacting proteins were mainly derived from *VvAGL3*, *VvAGL11*, *VvAGL62*, *VvAGL80*, and *VvAGL15*. Interestingly, *VvAGL6*, *VvAGL15*, *VvAGL62*, and *VvAGL80* are involved in both hormone-mediated GO and KEGG pathways (Table 1), suggesting that they can act as major factors in the regulation of hormone signalling pathways.

### 2.5. Screening of Hormone Cis-Elements in the Promoters of VvAGLs

To further understand the biological functions of the grape *AGL* gene family, cis-elements prediction revealed that this gene family mainly contains cis-elements related to hormone regulation and adversity stress, including the methyl jasmonate response element (TGACG-motif), abscisic acid response element (ABRE), salicylic acid response element (TCA-element), drought response element (MBS), and low-temperature response element (LTR). The hormone- and stress-responsive action components were quantified at the highest levels compared to the tissue-specific elements, with the exception of the light-responsive element, which had the highest number. This suggests that the *VvAGLs* may have a strong sensitivity to hormones and stress (Figure 6A). The fact that all members of the *VvAGL* family contain both the light-responsive element and the hormone-responsive element may mean these two cis-elements are critical for the growth and development of green plants. To further understand the potential roles of *VvAGL* family members in hormone response, we screened the hormone-responsive cis-elements in their promoters, and all members of the *VvAGL* family contained hormone-responsive cis-elements, mainly GA-, IAA-, SA-, ABA-, and MeJA-responsive cis-elements (Figure 6B), implying that most members of the grape *AGL* gene family may respond to multiple adversity stresses and hormone regulatory elements at the same time. In addition, all of them contain growth factors, light-responsive, and anaerobic-induced regulatory elements, which may be related to their corresponding regulation.

### 2.6. Differential Expression Profiles of VvAGLs and Their Interacting Proteins during Grape Seed Abortion Process

The differential expression profiles of 40 members of the *VvAGL* sub-family were analysed during the grape seed abortion using heat map analysis. As shown in Figure 7A, *VvAGL6-2, VvAGL11*, and *VvAGL62-11* exhibited high expression levels in the seed abortion, of which *VvAGL6-2* had the highest expression in the critical period of seed abortion, followed by *VvAGL11* and *VvAGL62-11*, which suggests that they may be the major factors in the grape seed abortion. Meanwhile, other members of the *VvAGL* sub-family were basically in the low expression. Interestingly, *VvAGL6-2* and *VvAGL62-11* had the highest expression levels in the second period of the seed abortion and lower expression in the other period, which indicated that *VvAGL6-2* and *VvAGL62-11* were the major regulatory factors in the critical period of grape seed abortion. In addition, *VvAGL11* exhibited a decreasing trend during seed abortion, indicating that *VvAGL11* might be one negative regulator in the whole process of seed abortion. Overall, *VvAGL6-2, VvAGL62-11*, and *VvAGL11* are the major factors of *VvAGLs*, playing a vital role during grape seed abortion. However, the expression of other genes of the *VvAGL* family was relatively low and did not change significantly, suggesting that they were involved in grape seed abortion but did not play a major regulatory role. More than half of the 40 *VvAGL* sub-family genes showed no change in expression during the process of grape seed abortion, implying that *VvAGL* family members respond differently to the regulation of grape seed abortion.

Based on the expression profiles of genes interacting with *VvAGLs* (Figure 7B,C), we screened the expression profiles of *AGL*-interacting genes in grape seeds at different developmental periods, and the results showed that 32 genes such as *VIT_206s0004g01910* (*VvANT*), *VIT_203s0038g01930* (*VvPCKR1*), *VIT_218s0001g09850* (*VvMYB44*), and *VIT_215s0001g05250* (*VvPAP2-4*), which were highly expressed in the early period of seed abortion. Among them, the expression of 16 genes, such as *VvANT*, *VvPCKR1*, *VvMYB44*, and *VvSPL12* decreased significantly, which was similar to the results of *VvAGL11*. Thus, these genes might enhance *AGL* family signalling during the regulation of seed abortion. Additionally, *VvANT* was predicted to interact with *VvAGL3* and *VvAGL15*. Similarly, both *VvMYB44* and *VvPCKR1* interact with *VvAGL15*, *VvAGL62*, and *VvAGL80*, while *VvSPL8* is associated with *VvAGL11*. Thus, these interacting genes might also mediate *VvAGL* gene expression and regulate grape seed abortion. Meanwhile, we observed another 16 genes interacting with *VvAGLs* with a significant upward trend in expression levels during grape seed abortion, *VIT_204s0044g00580* (*VvACT7*), *VIT_208s0040g00520* (*VvSERK1*), *VIT_218s0001g14400* (*VvPCKR1*), *VIT_212s0059g01380* (*VvACO3*), and other genes had significantly higher expression in the late period of seed abortion, which displayed a differential trend with *VvAGLs*, suggesting that *VvAGLs* might be mediated the grape seed abortion with the interacting protein as different modes of expression. Among them, the expression of *VIT_201s0010g02460* (*VvGAPC2*) and *VIT_217s0000g00830* (*VvSWEET10*) genes increased more than 10-fold. These genes are thought to interact mainly with *VvAGL15*, *VvAGL6-1*, and *VvAGL18*, which implies that, in the *VvAGL* family, they may be the main regulators inducing the grape seed abortion.

### 2.7. Validation of VvAGLs Expressions Using RT-qPCR Analysis

To validate expression patterns obtained from RNA-Seq, six genes (i.e., *VvAGL11, VvAGL80, VvAGL6-2, VvAGL3, VvAGL15*, and *VvAGL62*) that were the core genes in *VvAGLs* and highly expressed in grape seed abortion were selected to validate using RT-qPCR analysis (Figure 8). The expression profiles of RT-qPCR are similar to those from RNA-Seq, supporting RNA-Seq data that could represent genes’ relative expression levels at the different development stages in grape seed. The relative expression levels of *VvAGL6-2, VvAGL3, VvAGL15, VvAGL11, VvAGL80*, and *VvAGL62* were high among *VvAGLs* sub-family that demonstrated an overall decreasing trend in the seedless varieties, and only the expression of *VvAGL15* and *VvAGL6-2* showed an increasing trend, and they exhibit different trends in the seeded variety species. Respectively, *VvAGL11* showed higher expression in seeded variety species and lower expression in the seedless varieties. As Figure 8 shows, the six genes exhibited different expressions, *AGL11/AGL62* gradually increased in seed development and decreased in abortive development, whereas *AGL80/AGL6-2* increased with abortive development and remained almost unchanged in seed, and the pattern of *AGL3* in seeded/seedless development was less significant. In addition, both data yielded similar results, differing only at 30 days, which may stem from differences in the early period of fruit development in the three grape varieties at different maturity stages.

### 2.8. Regulatory Network of VvAGLs in Response to Multi-Hormone Signals

In this work, we identified 40 *VvAGLs* and predicted 1069 potentially interacting genes with *VvAGLs* and 5 *VvAGL* members (*VvAGL6/62/11/15/80*) as the major factors of the AGL sub-family contributed to the hormone signalling pathway and metabolic pathway either by themselves or by interacting genes (Figure 9A). Moreover, among VvAGLs and interacting genes mentioned above, 23 are involved in phytohormone signalling (ko04626), including IAA, CTK, SA, ABA, and ethylene hormone signalling pathways (Figure 9B,C). From the genes involved in the ABA metabolic pathway, we screened for a differentially expressed gene interacting with VvAGLs, in which the expression of *VvNCED1-2*, *VvRABF1*, and *VvABIL2* was down-regulated, suggesting that they may have a mutually synergistic role with *VvAGLs* in ABA signalling; *VvPYL8*, *VvNCED1-1*, *VvABF2*, *VvPP2CA*, *VvRABA2A*, and *VvPYL9* were up-regulated in the expression of *VvNCED* as a key ABA synthase, mediating seed abortion by promoting synthesis. Similarly, *VvYUC10*, a key synthase in the IAA metabolic pathway, was down-regulated, which had an opposite effect between *VvAGL6-2* and *VvAGL15* with an obvious trend on the increase. Moreover, the corresponding interacting gene, *VvTIR1* (an important repressor in the growth hormone signalling pathway), was decreased, from which we deduced that *VvYUC10* and *VvTIR1* might interact with *VvAGL6-2* and *VvAGL15* to involve seed abortion in a different expression, thereby promoting IAA signalling. Decreased expression of *VvAHK3* inhibited the expression of *VvARR3*, a key enzyme gene in CTK synthesis; meanwhile, raising the expression of *VvCTR* might promote the expression of *VvACO* and *VvEIN3* genes, as the key enzyme genes in ETH synthesis, which implies that CTK and ETH hormone metabolism has an antagonistic effect in the seed abortion. Further analysis revealed that both *VvAGL62* and *VvAGL11* may be involved in the regulation of the SA signalling pathway by interacting with the *VvTGA2/4* genes and may be antagonistic to each other and that *VvPR-1*, as a gateway repressor in the SA signalling pathway, maintains the dynamic balance of SA signalling by down-regulating *VvTGA2/4* and *VvNPR1/3*. From these results, we found that VvAGL family members may interact with each other to affect grape seed development by participating in different hormone signalling pathways, and some of them may participate in two or more signalling pathways, and VvAGLs, as the core transfer family, participate in multiple hormone pathways; meanwhile, *VvAGL6*, *VvAGL11*, *VvAGL15*, *VvAGL62*, and *VvAGL80* are the key regulatory nodes mediating the multi-hormone signalling network in grape seed development.

## 3. Discussion

With the continuous development of bioinformatics technology and the improvement of sequencing technology [32], it has become possible to identify and characterise a certain gene family and the regulatory network at the genome-wide level. AGL genes, as a sub-family of *MADS-box* genes encoding important transcription factors, are involved in multiple processes of plant growth and development [33]. Based on the transcriptomic data, we identified 40 *AGL* genes in grapes, the *VvAGL* gene sub-family has a highly conserved MADS-MEF2-like structural domain, MADS structural domains, and K-frames [34]. Moreover, the structural distribution and lengths of exons and introns of the *VvAGL* family members are comparable to those of the homologous genes in *Arabidopsis thaliana* [14], rice [35], and tomato [36] are highly conserved in structure distribution and length. The promoters of *VvAGLs* contain cis-acting elements in response to hormones, suggesting that it may affect plant growth and developmental processes by responding to hormone synthesis and metabolism, similar to previous reports on the regulation of flower development, formation, and flowering in plants [37,38]. We revealed that *VvAGLs* and their interacting proteins might regulate grape seed abortion through differential expression in response to different hormone synthesis and metabolism pathways, supporting the promoters’ analysis above.

Among VvAGL sub-families, *VvAGL11* can mediate grape seed abortion by negatively regulating expression. Similarly, down-regulation of the *AGL11* gene in tomatoes resulted in seedless tomato fruits [39]. In *Arabidopsis*, *AtAGL11* had been demonstrated as a regulator to control the structure and development of seed coat [40]. In tomato (*Solanum lycopersicum*), the silencing of *SlAGL11* produced seedless fruit [16]. *AGL11* regulated ovule development and ectopic expression of *AGL11* induced the formation of ovules on the sepals and petals in petunia [41]. The ectopic expression of *LlMADS2*, an *AGL11* homolog from lily (*Lilium longiflorum*), caused the conversion of sepals and petals to carpel and stamen-like structures in transgenic *Arabidopsis* [42]. Conversely, heterologous ectopic expression of *AGL11* in *Arabidopsis* and rice [38] could not induce ectopic ovule formation. *AGL11* also played a role in flowering regulation, and the over-expression homologous gene of *AGL11* caused early flowering in lily [42]. In grapes, during the development of ovules, the *VvAGL11* expression was maximal during the final stages of ripening, and, after the fruit set, expression was significantly reduced in seedless grapes [12], which was consistent with our findings. These results indicated that *AGL11* might function in a species-dependent mode and exhibit functional diversity across various plant species. In addition, other genes of the AGL sub-family also played different roles in the growth and development of diverse plants. For example, *VvAGL14* was involved in the regulation of the cell cycle of root meristematic tissue and promoted the expression of the growth hormone transcription factor to regulate root development [43]. The high expression of *AGL12* in rice induced the up-regulation of jasmonate, ethylene, and reactive oxygen species biosynthetic genes [44]. *AGL17* and *AGL12* were highly expressed in rice roots and involved in the regulation of root development in response to nitrate [45]. The loss of *AGL23* function leads to embryo abortion [46]. *AGL79* expression could be involved in the regulation of lateral root growth in plants [47]. *AGL36* was involved in the regulation of endosperm development [20]. *AGL17*, *AGL19*, and *AGL24* were involved in the regulation of flowering via photoperiod [45,48,49].

Numerous studies have reported that MADS-box transcription factors were involved in regulating the process of plant seed development and play an important role in flower development, control of flowering time in flowering plants, and partially in the development of plant nutrient organs [11]. Our proposal that VvAGL might regulate grape seed abortion by responding to multiple hormonal interactions is supported by this work, in which we identified all members of the *VvAGL* family and formed the crosstalk of major *VvAGLs* to describe above and revealed their potential regulatory network in response to multi-hormonal interactions to regulate grape seed abortion (Figure 10). Moreover, we found that only several *VvAGLs* were regulated in seed abortion by participating in hormone synthesis and signal transduction; however, most VvAGL members remained almost unchanged in this process, and the relevant genes in the ABA, CTK, SA, ETH, IAA signalling pathway were altered with changes in *AGLs* expression during grape seed abortion, of which *AGL11*, *AGL80*, *AGL62*, *AGL15*, and *AGL6-2* may have a critical role in plant seed abortion by responding to different hormonal changes. Similarly, the previous reports that *CmAGL11* plays an important role in the somatic embryogenesis process in Chinese chestnut, possibly by regulating GA, IAA, and ETH pathways [50]; in tomato, the genome-wide transcriptomic profiling of *Sl-AGL11*-overexpressed fruit and sepals also showed that the expression of genes related to GA, IAA, and ETH pathways was changed by *Sl-AGL11* overexpression [22]. In *Zanthoxylum armatum*, *AGL11* was involved in the biosynthesis and signalling of ABA, ETH, CTK, GA, IAA, and SA [24]. Transcriptome expression profiling of *Jatropha curcas* ovule development revealed that AGL regulates ovule development by upregulating the expression levels of genes related to signalling pathways in response to GA and JA signalling processes; moreover, the upregulation of BR signalling genes during ovule development might have been regulated by other phytohormone signalling pathways through crosstalk [51]. The expression level of AGL in the seed of grapevine saprophytes exogenously administered with GA_3_ showed an initial increase and then a decrease, indicating that AGL can mediate the seed abortion process by responding to the hormone change [52]. Recent molecular analyses illustrated how GA treatment induced parthenocarpy by influencing the miRNAs and GAMYB genes [53]. As seed development progressed, the GAMYB gene was increasingly expressed in seeded grapes, and the GAMYB gene response indicated that GA functions in seed development and seed abortion. Indeed, gibberellic acid (GA) and abscisic acid (ABA) were commonly used to increase berry size, reduce seed trace size, and promote seed development [54,55]. The cis-element analysis of the *VvAGL11* promoter in ‘Flame Seedless’ treated with GA3 showed that *VvAGL11* was up-regulated, which suggests that exogenous GA3 could affect seed coat development [56]. Furthermore, the transcript levels of the salicylic acid (SA) genes were higher in ‘Thompson Seedless’ [56]. All of these results suggest that AGL plays an important role in modulating hormone balance. Besides these, AGLs interacted with five hormone-responsive transcription factor genes, thereby leading to seed abortion, indicating that VvAGLs promote the induction of seed abortion, possibly by mediating multi-hormonal signalling pathways.

As described above, although the previous studies confirmed that *AGL11* is the important regulator of plant seed abortion [13], and other several AGL members were also involved in the modulation of this process [57], little is known AGL sub-family-mediated regulatory network through responding to multi-hormone signals during seed abortion in plants including grapes. Here, we identified the differential expression profiles of VvAGL sub-family in the grape seed abortion process, and revealed that several VvAGLs might be involved in the modulation of this process by potentially interacting with genes in various hormone signals; meanwhile, we developed one potential regulatory network of VvAGLs to modulate the grape seed abortion process through the mediation of the multi-hormone signal pathways at the whole transcriptome level. Our findings provide new insights into the AGL gene-mediated regulation of plant seed abortion, which could be helpful in the molecular breeding of the seedless grapes in the table for the wine and grape juice industry.

## 4. Materials and Methods

### 4.1. Plant Materials

Six-year-old European and American hybrid diploid grapes ‘Zhengyan seedless’ (*Vitis vinifera* × *V. labrusca*) grown in Nanjing, China (N 32°02′12.77″, E 118°37′33.25″), were chosen as the experimental material. The plant materials were grown under common field conditions at the grape experimental base of Jiangsu Academy of Agricultural Sciences, Lishui, China (our long-term partners). Seed and berry samples were collected randomly from different branches at different stages: seed development stage [25 days after flowering (25DAF), 30 days after flowering (30DAF), 40 days after flowering (40DAF)], early stage of abortion [48 days after flowering (48DAF)], late stage of abortion [60 days after flowering (60DAF), 80 days after flowering (80DAF)]. All collected samples were snap-frozen in liquid nitrogen and stored at −80 °C for further analysis.

### 4.2. Measurement of Fruit Physiological Indexes

The horizontal and vertical lengths of the berries and seeds were measured using an automatic vernier calliper. The experimental data were analysed using Excel, PowerPoint 2019, and GraphPad Prism 8.0.

### 4.3. Phylogenetic Analysis

The software MEGA version 11.0 and ClustalX2 were employed to conduct the phylogenetic analysis. The theoretical gene and protein sequences of the grape *VvAGLs* family were obtained from the CRIBI database (http://genomes.cribi.unipd.it/grape/index.php accessed on 8 August 2023). We constructed the phylogenetic tree using the Maximum Likelihood method against the target species genome and the reference genome. The genes in gene families that were single copies in the whole genome of sequencing species and reference species were used, and the evolutionary tree was constructed using MEGA11.0 to study the evolutionary relationship between species. Based on the results of the clustering analysis of a homologous gene family, a single copy of a homologous gene was selected for multi-sequence alignment (using MUSCLE software for sequence alignment), and then a phylogenetic tree was constructed based on the single-copy gene method. Similarly, one more phylogenetic tree was prepared using all the available *AGL* protein sequences from grapes and the other five species (*Arabidopsis*, orange, apple, corn, and tomato). The physical and chemical characterisation of the *AGL* family genes was performed using an online website ProtParam (https://web.expasy.org/protparam/, accessed on 10 August 2023), and the parameters obtained were represented in bar charts.

### 4.4. Prediction of VvAGL Gene Intron–Exon Structure

The exon/intron structure of the *VvAGL* genes was determined by comparing the predicted coding sequence and its corresponding genomic sequence using GSDS software (http://gsds.cbi.pku.edu.cn, accessed on 11 August 2023).

### 4.5. Analysis of Cis-Acting Elements in the Promoter Region of VvAGLs

Sequences 1500 bp upstream of the VvAGL transcription start site (ATG) were analysed from the grapevine genome database CRIBI (http://genomes.cribi.unipd.it/grape/, accessed on 12 August 2023) and the Grapevine Genome Browser (http://www.genoscope.cns.fr/externe/GenomeBrowser/Vitis/, accessed on 12 August 2023) The PLANTCARE online database (http://bioinformatics.psb.ugent.be/webtools/plantcare/html/, accessed on 13 August 2023) was used to search for cis-acting elements in the promoter region of VvAGLs.

### 4.6. RNA Extraction, Library Construction, and Sequencing

Total RNA was extracted from different samples of control grape ovaries at 25, 32, 40, 48, 60, and 80 days using our modified CTAB method [58]. RNA quality was assessed on an Agilent 2100 Bioanalyzer (Agilent Technologies, Palo Alto, CA, USA) and RNAase-free agarose gelelectrophoresis was used for Examination. After extraction of total RNA, eukaryotic mRNA was enriched with Oligo (dT) magnetic beads. Enriched mRNA was then fragmented into short fragments using fragmentation buffer and reverse transcribed into short fragments by using the NEBNext Ultra RNA Library Preparation Kit for Illumina (NEB#7530, New England Biolabs, Ipswich, MA, USA) was reverse transcribed to cDNA. Purified double-stranded cDNA fragments were end-repaired, “A” bases were added, and ligated to the Illumina sequencing junction. Ligation reactants were purified with AMPure XP microbeads (1.0×). The ligated fragments were size-selected using agarose gel electrophoresis and amplified using polymerase chain reaction (PCR). The resulting cDNA library was sequenced using Illumina Novaseq6000 from Gene Denovo Biotechnology Co. (Guangzhou, China).

### 4.7. Analysis of Differentially Expressed Genes

Read count data were used as input data for the analysis of differentially expressed genes in the software package DESeq 2. The analysis was divided into three parts: (1) normalising read counts, (2) calculating hypothesis testing probabilities (*p*-values) based on the model, and (3) performing multiple hypothesis tests and corrections to obtain the false discovery rate (FDR). Based on this analysis, FDR < 0.01 and |log2FC| ≥ 1 (where FC is the fold change) were screened as significantly different and categorised as differentially expressed genes (DEG). Transcripts |log2FC| < 0.25 were considered to have no change in expression level. Other transcripts (0.25 |log2FC| < 1) were considered “slightly upregulated” or “slightly downregulated”.

### 4.8. Functional Annotation and Enrichment Analysis of VvAGL Differentially Expressed Genes

Using the Kyoto Encyclopedia of Genes and Genomes database (KEGG) (https://www.kegg.jp/, accessed on 14 August 2023 ) and Gene Ontology (GO) (http//geneontology.org/, accessed on 14 August 2023), we performed enrichment analysis and deciphered the interaction network of *VvAGL.* Heatmap analysis was performed using TB tools v2.056.

### 4.9. Prediction of Interacting Proteins

Based on our RNA-seq data, we used the online software STRING v10.0 (http://string-db.org, accessed on 15 August 2023) to predict the interacting proteins of the VvAGL family. For this analysis, we performed the following: select “Multiple proteins” and enter the gene numbers of the VvAGL family members in “Protein name”; under the term “Organism”, enter the Vitaceae family; and then, for VvAGL family members, enter the gene numbers of the VvAGL family members in “Protein name”. Under “Organisms”, we entered the family *Vitis vinifera*, and then we predicted the results for the VvAGL-interacting proteins.

### 4.10. Quantitative Real-Time Polymerase Chain Reaction (qRT-PCR) Analysis

To evaluate the reliability of the sequencing results, 6 VvAGLs were selected randomly for qRT-PCR verification. The actin gene was used as a reference gene in the qRT-PCR detection of VvAGLs (primer sequences are shown in Appendix A). The RNA of grape seeds was extracted using the CTAB method. For the amplification system and procedure, we used the SYBR^®^ Premix Ex TaqTM II kit from TaKaRa Company (Beijing, China) according to the instructions. All cDNAs were stored at −80 °C for the next analysis. All experiments were performed with 3 biological replicates, and relative expression was calculated according to the 2^−∆∆ct^ method.

## 5. Conclusions

In this study, we present the identification and characterisation of the VvAGLs genes and their involvement in the regulation of grapevine seed abortion in response to a variety of GA-mediated hormone signalling. The network of VvAGLs involved in the regulation of grapevine embryo failure was obtained through gene family identification, reciprocal protein prediction, and promoter action element analysis. Our findings suggest that VvAGLs might negatively modulate grape seed abortion through mediating multi-hormones, which could provide important implications for the molecular breeding of seedless grapes.

## Figures and Tables

**Figure 1 ijms-25-09849-f001:**
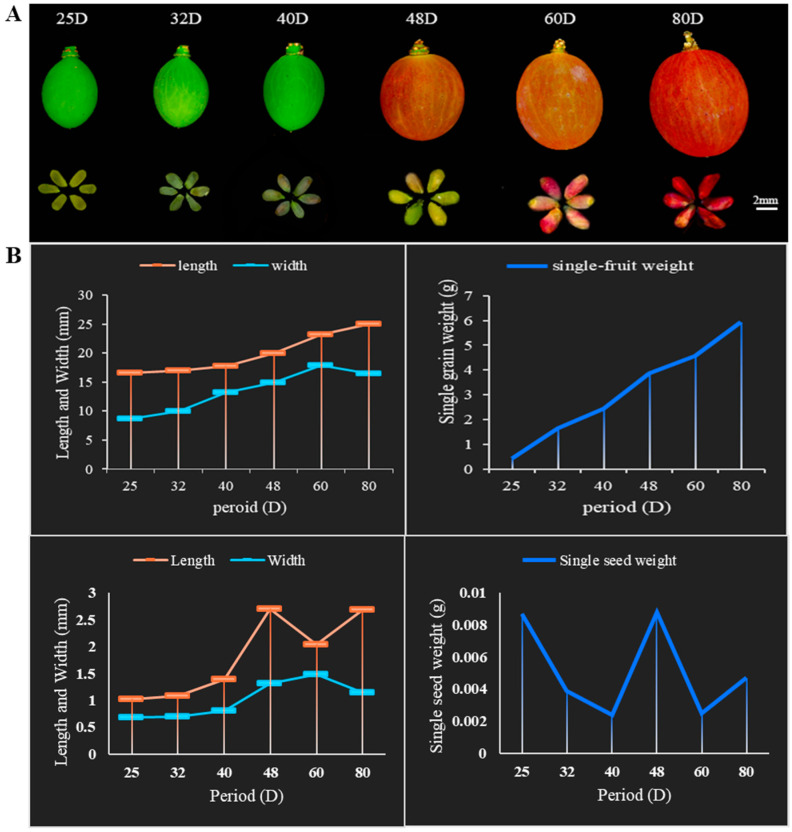
Morphological changes in berries’ development during grapevine seed abortion process: (**A**) Morphological changes of ‘Zhengyan seedless’ during berries and seeds development in seeds abortion process. (**B**) The length and width of grapefruits in the seed abortion process.

**Figure 2 ijms-25-09849-f002:**
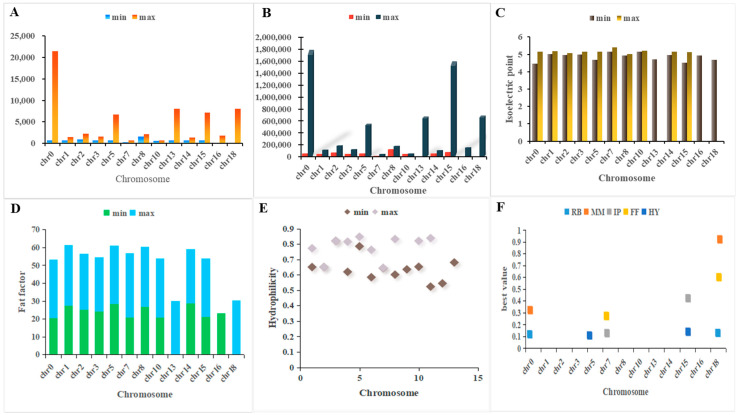
Physicochemical properties of AGL family genes: (**A**) Residual base. (**B**) Molecular mass/KD. (**C**) Isoelectric point. (**D**) Fat factor. (**E**) Hydrophilicity. (**F**) Maximum distribution of chromosomes. The X-axis indicates the chromosome on which the AGL sub-family genes are located, and the Y-axis indicates the maximum and minimum values of the corresponding physicochemical properties of the genes distributed on this chromosome.

**Figure 3 ijms-25-09849-f003:**
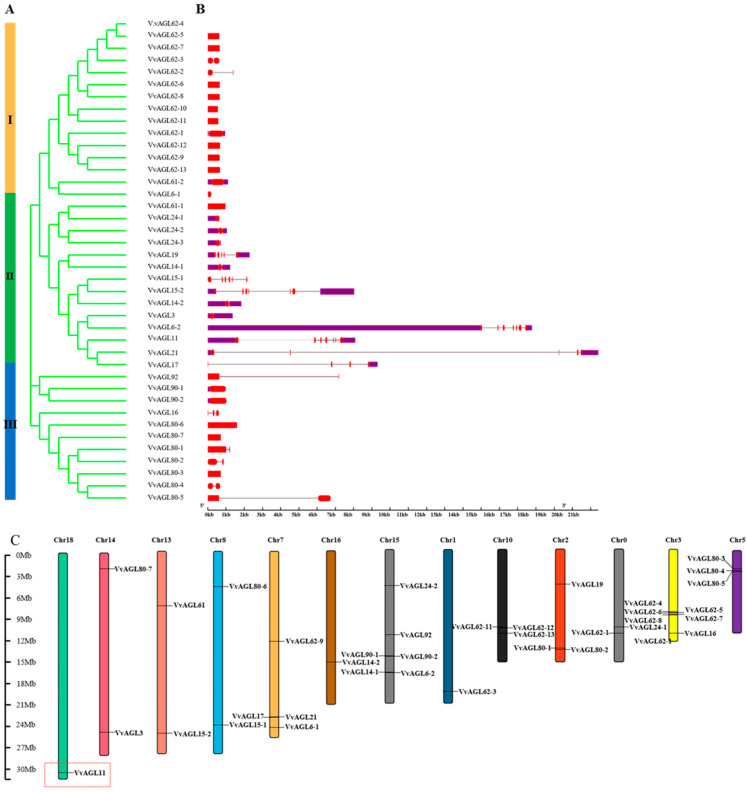
Sequence analysis and chromosomal location of *VvAGLs*: (**A**) The phylogenetic tree generated using the MEGA11.0 program with the Maximum Likelihood method. Yellow, green and blue colours represent different branches; red colour indicates the CDS region of the gene structure and purple colour represents the exons of the gene. (**B**) The exon–intron composition of *AGL* genes. The coding sequences (CDS) and up- or down-stream regions of *AGL* genes are represented by red and purple boxes, respectively. Lines represent the introns. (**C**) The chromosome localisation of *VvAGLs*. The red dashed box indicates the distribution of key AGL family members on the chromosome. Different colours represent different chromosomes.

**Figure 4 ijms-25-09849-f004:**
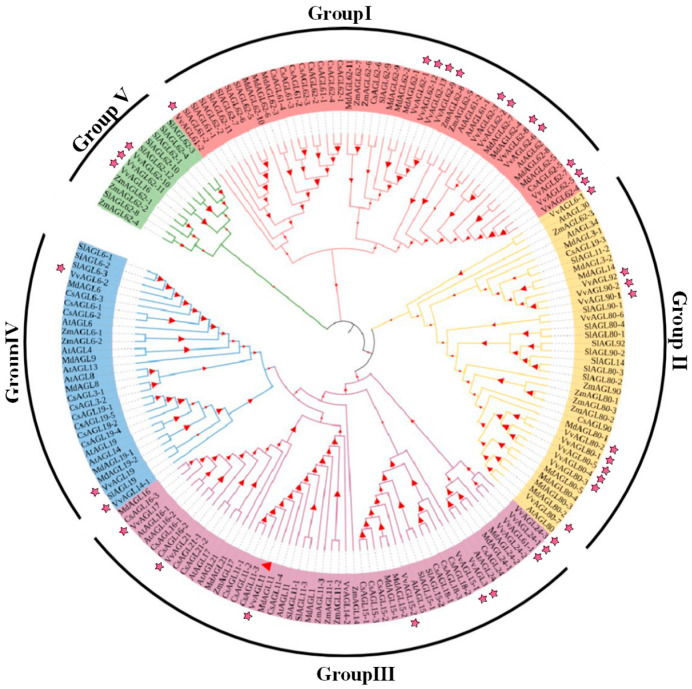
Phylogenetic analysis of the AGL family across six species: The phylogenetic tree was generated after an alignment of deduced *Vitis vinifera*, *Zea mays*, *Arabidopsis thaliana*, *Citrus sinensis*, *Malus domestic*, and *Solanum lycopersicum AGL* domains at the N-terminus. We constructed the phylogenetic tree using the Maximum Likelihood method against MEGA11.0 to study the evolutionary relationship between species. Based on the results of the clustering analysis of the homologous gene family, a single copy of the homologous gene was selected for multi-sequence alignment [using MUSCLE software (MEGA11.0) for sequence alignment] and a phylogenetic tree was constructed based on the single-copy gene method. Stars represented VvAGLs family members, triangular stars represented the main members of VvAGLs, different colors represented different groupings.

**Figure 5 ijms-25-09849-f005:**
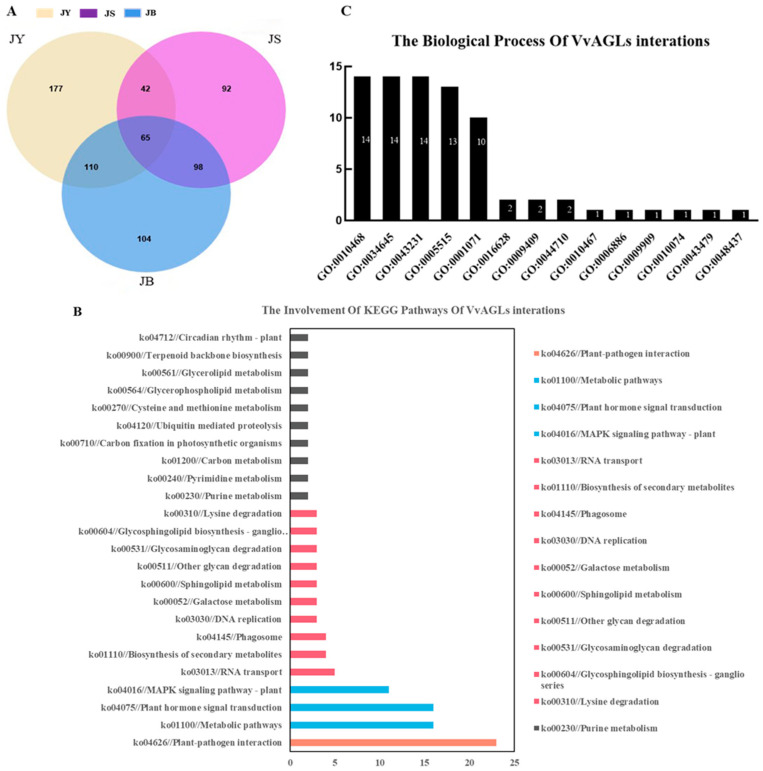
GO and KEEG enrichment analysis of *VvAGLs* family: (**A**) Number of interacting proteins in different periods. (**B**) The KEGG pathway annotation of the top 42 significant enriched pathways. (**C**) The GO analysis of 14 biological processes. Numbers on columns: The number of interacted genes involved in the GO pathway. ko00604//Glycosphingolipid biosynthesis—ganglio series.

**Figure 6 ijms-25-09849-f006:**
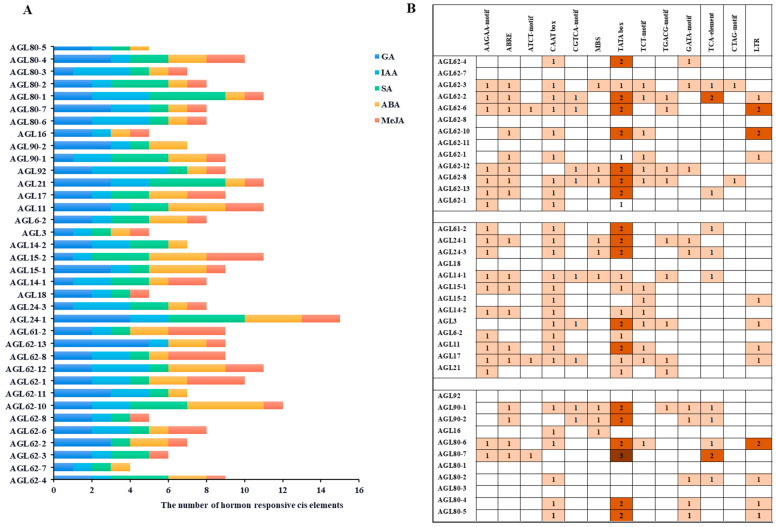
Screening and analysis of cis-elements in the promoters of *VvAGLs*: (**A**) The number of hormone-responsive cis-elements in promoter regions of *VvAGLs*; (**B**) Types of hormone-responsive cis-elements in promoter regions of *VvAGLs.* ABRE: ABA-responsive cis-elements; AuxRR-core and TGA-element: IAA-responsive cis-elements; CGTCA-motif and TGACG-motif: MeJA-responsive cis-elements; TCA-element and SARE: SA-responsive cis-elements; GARE-motif, P-box and TATC-box: GA-responsive cis-elements. ABA: abscisic acid; CTK: cytokinin; AUX/IAA: auxin–response repressor protein/indole acetic acid; SA: salicylic acid; MeJA: methyl jasmonate. Lighter to darker colous represented an increase in the number of cis-elements.

**Figure 7 ijms-25-09849-f007:**
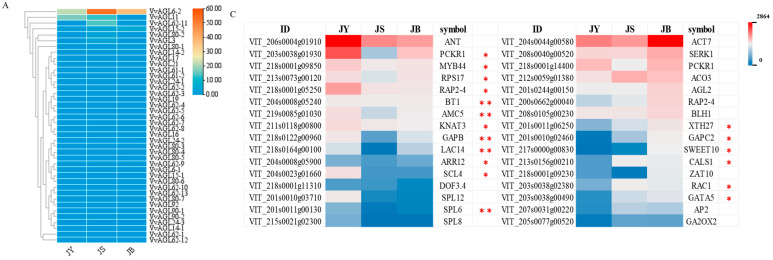
Expression analysis of *VvAGL* and reciprocal proteins: (**A**) Expression of the *VvAGL* family at different developmental periods. (**B**) Expression of *VvAGL* reciprocal proteins at different developmental periods. (**C**) Proteins are specifically expressed in different hormone metabolic pathways. JY: Stage of seed development; JS: Stage of seed beginning to abort; JB: Late stage of seed abortion; Asterisks indicate a significant difference between JY, JS, and JB by Student (* *p* < 0.05; ** *p* < 0.01).

**Figure 8 ijms-25-09849-f008:**
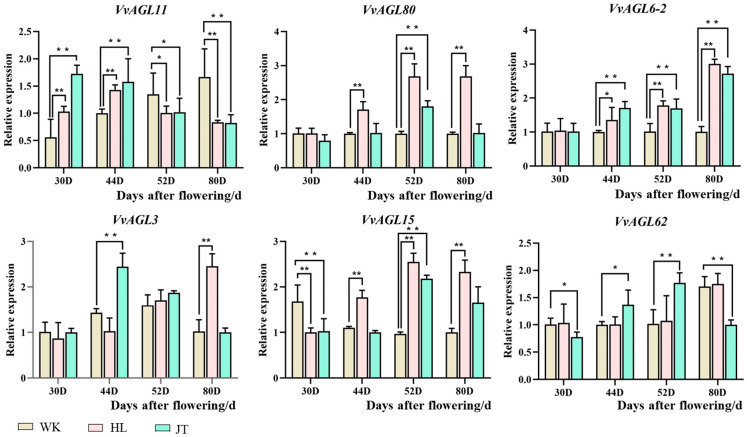
Expression of *VvAGL* family members in grape seed. Asterisks indicate a significant difference between different development periods using Student’s *t*-test (* *p* < 0.01; ** *p* < 0.05); WK ‘Wink’; HL ‘Blush seedless’; JT ‘Jintian Huangjia seedless’.

**Figure 9 ijms-25-09849-f009:**
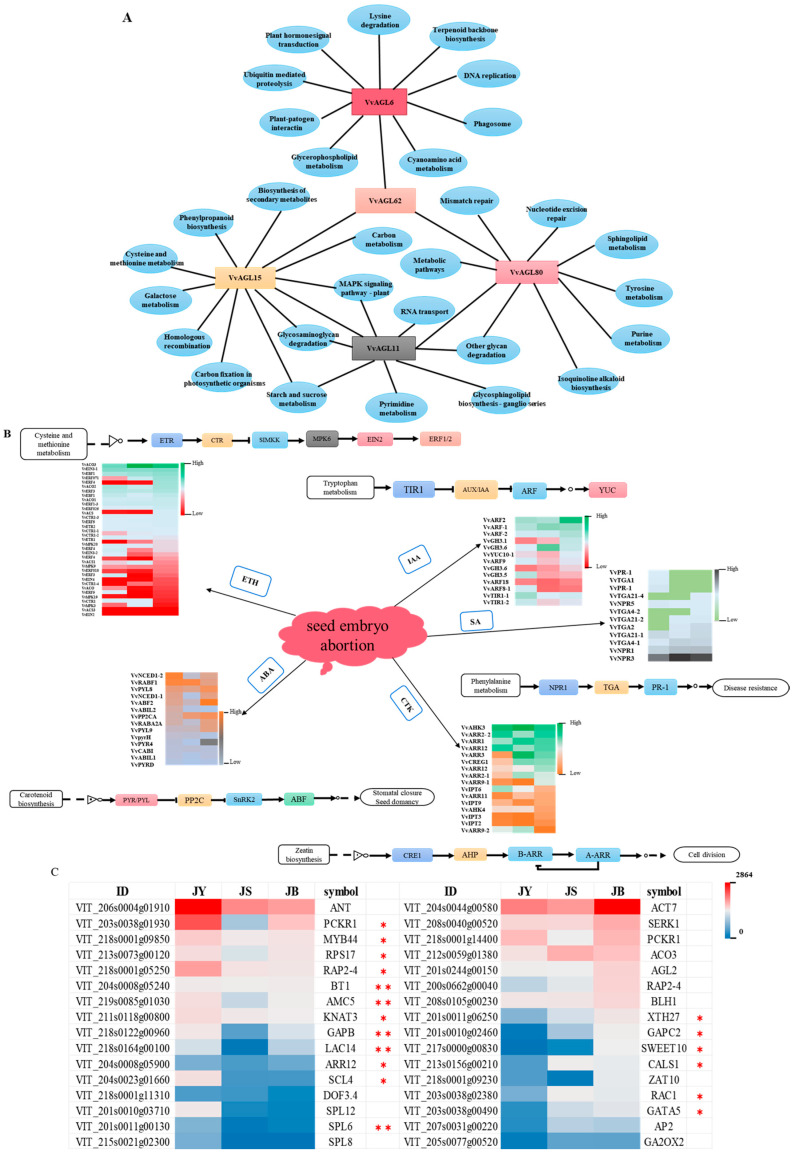
VvAGL-interacting genes and polyhormonal regulatory networks: (**A**) AGL family genes and interacting proteins are involved in the metabolic processes. (**B**) Involvement of VvAGL-interacting genes in the polyhormonal interaction network. (**C**) The levels of VvAGLs expression and the involvement of hormonal biosynthesis and signalling genes in plant growth and development. Asterisks indicate a significant difference between different development periods using Student’s *t*-test (* *p* < 0.01; ** *p* < 0.05).

**Figure 10 ijms-25-09849-f010:**
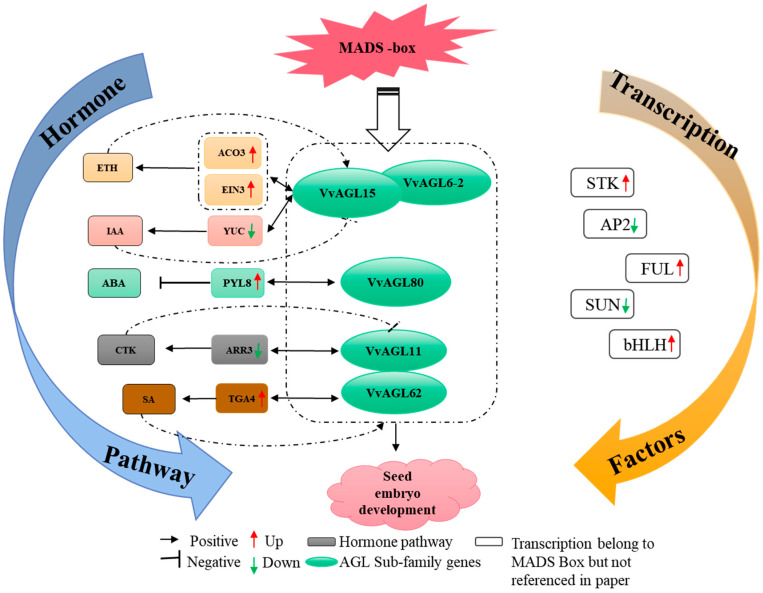
Schematic representation of plant hormone metabolism and signalling pathway gene network during grape seed abortion. The genes involved in the CTK signalling pathway are represented by a grey box; the genes involved in the SA signalling pathway are represented by a brown box; the genes involved in the IAA signalling pathway are represented by an orange box; the genes involved in ETH signalling pathway are represented by the yellow box; the genes involved in ABA signalling pathway are represented by the green box. STK, AP2, FUL, SUN, and bHLH were referenced in previous studies but not identified in this study. The bidirectional arrows represented the interaction between the two genes.

**Table 1 ijms-25-09849-t001:** GO and KEGG pathway of *VvAGLs*.

Gene ID	Symbol	Pathway	K-ID	Go Process
VIT_200s0250g00085	AGL61-1	ko04075//Plant hormone signal transduction	K14486	GO:0010468//regulation of gene expression; GO:0034645//cellular macromolecule biosynthetic processGO:0010467//gene expression
VIT_207s0129g00650	AGL6-1			
VIT_200s1450g00005	AGL62-1			
VIT_201s0010g01500	AGL62-2			
VIT_203s0088g00510	AGL62-4			
VIT_203s0088g00550	AGL62-5			
VIT_203s0088g00590	AGL62-6			
VIT_203s0088g00600	AGL62-7			
VIT_203s0088g00610	AGL62-8			
VIT_207s0005g06590	AGL62-9			
VIT_210s0003g03015	AGL62-10			
VIT_210s0003g03020	AGL62-11			
VIT_210s0003g03025	AGL62-12			
VIT_210s0003g03970	AGL62-13			
VIT_200s0211g00110	AGL17	ko04075//Plant hormone signal transduction	K14486	GO:0010468//regulation of gene expression;GO:0034645//cellular macromolecule biosynthetic processGO:0044710//single-organism metabolic processGO:0006886//intracellular protein transport;GO:0009409//response to cold;GO:0009909//regulation of flower development;GO:0010074//maintenance of meristem identityGO:0043479//pigment accumulation in tissues in response to UV light;GO:0048437//floral organ developmentGO:0000904//cell morphogenesis involved in differentiation;GO:0009664//plant-type cell wall organization;GO:0009888//tissue developmentGO:0048580//regulation of post-embryonic development;GO:0080154//regulation of fertilization
VIT_200s0211g00180	AGL21			
VIT_200s0250g00085	AGL61-2			
VIT_200s0729g00010	AGL24-1			
VIT_202s0025g04650	AGL19			
VIT_203s0167g00100	AGL24-2			
VIT_208s0007g08790	AGL15-1			
VIT_213s0158g00100	AGL15-2			
VIT_214s0068g01800	AGL3			
VIT_215s0024g02000	AGL24-3			
VIT_215s0048g01240	AGL14-1			
VIT_215s0048g01270	AGL6-2			
VIT_216s0022g02400	AGL14-2			
VIT_218s0041g01880	AGL11			
VIT_202s0109g00382	AGL80-6	ko04075//Plant hormone signal transduction	K14486	GO:0010468//regulation of gene expression; GO:0034645//cellular macromolecule biosynthetic process
VIT_202s0109g00384	AGL80-7			
VIT_203s0097g00192	AGL16			
VIT_205s0020g01043	AGL80-1			
VIT_205s0020g01046	AGL80-2			
VIT_205s0020g01055	AGL80-3			
VIT_208s0032g00974	AGL80-4			
VIT_214s0060g00300	AGL80-5			
VIT_215s0021g00560	AGL92			
VIT_215s0021g02220	AGL90-1			
VIT_215s0021g02250	AGL90-2			

## Data Availability

The RNA-seq data have been deposited into the NCBI under the accession number PRJNA1156312 (https://www.ncbi.nlm.nih.gov/bioproject/PRJNA1156312, accessed on 15 August 2023). All data generated or analysed during this study are included in this published article and its Appendix A.

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
