# Peer review of "Identification of VvAGL Genes Reveals Their Network’s Involvement in the Modulation of Seed Abortion via Responding Multi-Hormone Signals in Grapevines"

_ijms, 2024, doi:10.3390/ijms25189849_

Round 1

Reviewer 1 Report (New Reviewer)

Comments and Suggestions for Authors

The manuscript entitled "Identification and characterization of VvAGL genes reveals their network involved in the modulation of seed embryo abortion via responding to GA-mediated multi-hormone signals in grapevines" is an interesting report, in which authors identified and characterized VvAGL genes at grape transcriptome-wide level and elucidated the network and key target site genes of their modulating grape seed embryo abortion in responsive to GA-mediated multi-hormone signals. The study provides comprehensive insights into the roles of VvAGLs mediating multi-hormone signal pathways during modulation of grape seed abortion process, and contributes significantly to further recognizing the molecular mechanism of seed abortion and provides the critical target site gene resource for molecular breeding of seedless grape with high quality. In addition, it appears that there are several points, listed below, that need to be addressed by the authors before acceptance.

1. The title of this article is too long, and authors should make it simply.

2. In the "Introduction" section, suggest that the research gap be more clearly defined by summarizing the current state of knowledge and then explicitly stating what is expected to be further done about the AGL-mediated seed abortion in plant species, especially, in grapes. This could include a more detailed explanation of how existing studies have not fully addressed the molecular mechanisms or genetic regulation involved.

3. As to seed abortion and seed embryo abortion, this is two different concepts, please carefully distinguish and accurately use them in the manuscript.

4. In 4.1 Sub-section, it might be useful to specify the cultivation conditions under which the plants were grown. Such details are critical for reproducibility.

5. In "Discussion", suggest expanding the discussion to include potential molecular mechanisms or signaling pathways through which AGLs might mediate plant seed abortion in response to multi-hormone signals, especially, in grapes. Consider including existing literature on seed abortion or propose hypotheses based on the observed gene expression patterns.

6. In plant materials, please state why to choose the 25, 32, 40, 48, 60 and 80 days after flowering to collect samples, which is important for elicidating the roles of VvAGLs during modulation of grape seed abortion.

Minor Comments

- Some figures, particularly those related to gene expression and morphological changes in berries, could benefit from higher resolution or additional labeling for clarity.

- As to transcription factors, please use their font format correctly, regular or italic font.

- References are comprehensive but could be updated to include the most recent studies in the field, enhancing the manuscript's relevance and context.

- Please carefully check and correct the reference list and citation in text, and uniform their formats.

- In the line of 323, the standard name of ‘Weike’ is ‘Wink’, the standard name of ‘Honglian seedless’ is ‘blush seedless’.

-in the line of 454, ‘Zhengyan seedless’ is Vitis vinifera×V.labrusca.

Author Response

The response to comments from Reviewer 1#

Q 1: The title of this article is too long, and authors should make it simply. 

Response 1: Thanks for your careful suggestion. Following your advice, we checked the title of this article and make it simply. The title was revised as “Identification of VvAGL genes reveals their network involved in the modulation of seed abortion via responding multi-hormone signals in grapevines” in the modified text.

Q 2: In the "Introduction" section, suggest that the research gap be more clearly defined by summarizing the current state of knowledge and then explicitly stating what is expected to be further done about the AGL-mediated seed abortion in plant species, especially, in grapes. This could include a more detailed explanation of how existing studies have not fully addressed the molecular mechanisms or genetic regulation involved.

Response 2: Thanks for your good suggestion. Based on your suggestions, we presented the more detail the current progress of research on seedless grape and AGLs and described the research gap in the molecular regulatory mechanisms addressed by the current research and provided a description of the research that will be carried out in the next steps.

Q 3: As to seed abortion and seed embryo abortion, this is two different concepts, please carefully distinguish and accurately use them in the manuscript.

Response 3: Thanks for your constructive suggestion. According to your suggestion, we carefully checked the whole article and use the suitable description “seed abortion” in the revised text.

Q 4: In 4.1 Sub-section, it might be useful to specify the cultivation conditions under which the plants were grown. Such details are critical for reproducibility.

Response 4: Thanks for your good suggestion. Following your advice, we have supplemented the related contents on the plant’s cultivation conditions in the 4.1sub-section.

Q 5: In "Discussion", suggest expanding the discussion to include potential molecular mechanisms or signaling pathways through which AGLs might mediate plant seed abortion in response to multi-hormone signals, especially, in grapes. Consider including existing literature on seed abortion or propose hypotheses based on the observed gene expression patterns.

Response 5: Thank you for your honest advice, based on your suggestions, we referenced to the new literature to add more research on AGLs mediating grape seed embryo abortion by responding to hormones signaling to further prove our findings and make the article completer and more logical.

Q 6: In plant materials, please state why to choose the 25, 32, 40, 48, 60 and 80 days after flowering to collect samples, which is important for elicidating the roles of VvAGLs during modulation of grape seed abortion.

Response 6: Thanks for your good suggestion. According to your suggestion, we carefully stated the reasons we choose samples at the 25, 32, 40, 48, 60 and 80 days after flowering in the plant materials.

Q 7: Some figures, particularly those related to gene expression and morphological changes in berries, could benefit from higher resolution or additional labeling for clarity.

Response 7: Thanks for your suggestion. Based on your advice, we had improved the resolution of the gene expression and morphological changes in berries for a better comprehension.

Q 8: As to transcription factors, please use their font format correctly, regular or italic font.

Response 8: Thanks for your good suggestion. Following your suggestion, we carefully checked the transcription factors in content and changed their font format correctly.

Q 9: References are comprehensive but could be updated to include the most recent studies in the field, enhancing the manuscript's relevance and context.

Response 9: Thanks for your constructive suggestion. According to your suggestion, we updated the most recent studies in the field to enhance the manuscript's relevance and context.

Q 10: Please carefully check and correct the reference list and citation in text, and uniform their formats.

Response 10: Thanks for your honest advice. According to your suggestion, we carefully checked and corrected the reference list and citation in text and uniformed their formats in the reference list and the revised text.

Q 11: In the line of 323, the standard name of ‘Weike’ is ‘Wink’, the standard name of ‘Honglian seedless’ is ‘blush seedless’.

Response 11: Thanks for your careful suggestion. Based to your suggestion, we had carefully revised and corrected them as the standard name of ‘Wink’ and‘blush seedless’ in the line of 323.

Q 12: in the line of 454, ‘Zhengyan seedless’ is Vitis vinifera×V.labrusca.

Response 12: Thanks for your careful suggestion. Following your advice, we had carefully corrected the related contens in the line of 454.

Reviewer 2 Report (New Reviewer)

Comments and Suggestions for Authors

The manuscript entitled ' Identification and characterization of VvAGL genes reveals 2 their network involved in the modulation of seed embryo abor-3 tion via responding to GA-mediated multi-hormone signals in 4 grapevines.* focuses on the identification and characterization of transcription factors of the MADS box family that are involved in the seed abortion process in seedles grapevine. 40 differentially expressed transcription factors were identified and characterized in detail. The authors established a network scheme for the complex interaction between transcription factors, and different genes of different hormone pathways. This is a very nice manuscript on a very interesting topic. I have a few minor suggestions:

Figure 2 needs more explanations. It remains unclear how the analysis was done (material and methods, and figure legend) and the relevance of all these data should be adressed in the result section.

Figure 6: in the legend abbreviations for GA, IAA, SA, ABA and McJA are missing. x axis should not be number of hormons but number of hormon responsive cis elements

Figure 7: legend should explain JY, JS and JB

Figure 8: I cannot see any astersks - so no significant differeences? y axis: day after flowing or flowering?

Figure 10: explain ll abbrevations of proteins and genes.

line 490: provide reference for your CTAB method.

line 529: qPCR primer sequences are missing

Author Response

The response to comments from Reviewer 2#

Q 1: Figure 2 needs more explanations. It remains unclear how the analysis was done (material and methods, and figure legend) and the relevance of all these data should be adressed in the result section.

Response 1: Thanks for your careful suggestion. Following your advice, we supplemented the analysis method about figure 2 and addressed all these data in the result section.

Q 2: Figure 6: in the legend abbreviations for GA, IAA, SA, ABA and MeJA are missing. x axis should not be number of hormons but number of hormon responsive cis elements.

Response 2: Thanks for your good suggestion. As suggested, we supplemented the related content of GA, IAA, SA, ABA and MeJA in the legend abbreviations, and modified the title of x axis in figure 6.

Q 3: Figure 7: legend should explain JY, JS and JB.

Response 3: Thanks for your constructive suggestion. According to your suggestion, we added the explanation of JY, JS and JB in figure 7.

Q 4: Figure 8: I cannot see any astersks - so no significant differeences? y axis: day after flowing or flowering?

Response 4: Thanks for your suggestion. Following your advice, we have checked chart content and added the asterisks and modified the title of the horizontal coordinate in the figure 8.

Q 5: Figure 10: explain ll abbrevations of proteins and genes. 

Response 5: Thanks for your good suggestion. Following your advice, we carefully checked the chart content and explained abbreviations of proteins and genes in the figure 10.

Q 6: line 490: provide reference for your CTAB method.

Response 6: Thanks for your suggestions. Based on your advice, we provided the reference for CTAB method in the revised text.

Q 7: line 529: qPCR primer sequences are missing.

Response 7: Thanks for your honest suggestions. Following your suggestion, we added the qPCR primer sequences in the supplementary table 2.

This manuscript is a resubmission of an earlier submission. The following is a list of the peer review reports and author responses from that submission.

Round 1

Reviewer 1 Report

Comments and Suggestions for Authors

The article focuses on the economic importance of grapevines (Vitis vinifera L.) and the increasing demand for seedless grapes in the food industry. It highlights the need to improve breeding techniques to develop new seedless grape varieties that meet market demands. Additionally, it describes the diversity of parthenocarpy in grapes and its relationship with seed abortion, underscoring the importance of understanding the underlying molecular mechanisms. The results reveal the identification and characterization of 40 AGL family genes in grapevines, as well as their structural and evolutionary conservation with AGL family members in other plant species. It is shown that these genes are associated with cis-regulatory elements responsive to hormones, suggesting their involvement in regulating plant growth and development in response to hormonal signals. The discussion delves into the role of AGL genes in regulating seed development in grapevines, emphasizing the importance of hormonal and genetic interactions in seed abortion. It is proposed that AGL genes, especially VvAGL11, may be crucial in this process, and it is suggested that hormonal regulation may be a significant factor in seed abortion in grapevines.

However, several minor revisions are suggested:

  -         Identify and correct any missing commas, periods, and italics throughout the document.

  -    Review references older than 5-10 years and update them with more recent sources. Eliminate references from older years (1994, 2000, 2004, 2008, 2003, 2002, 2006, 1995, 1997, 1998) to reflect current research in the field.

-          Significantly enhance figure resolution to improve readability. Address inconsistencies in figure formatting, such as font types, sizes, and background colors, to ensure visual cohesion across all figures.

-          The discussion section needs a clear and concise overview of the topic. I suggest considering the rephrasing of sentences to make them more concise and focused, maintaining direct links between the discussion and the results. Furthermore, reorganizing the information in a manner that creates a logical flow.

-          Enhance the conclusion by providing a comprehensive summary of findings and their implications. Elucidate tangible implications of the study's results beyond a mere recapitulation of the article's content. I would suggest to expand the discussion on the practical implications of the findings for the wine industry and agriculture in general.

-          Shorten the abstract summarizing the key points of the article without unnecessary verbosity.

-          Provide references for statements made in the paragraph of the introduction section “Based on the conditions of pollination and fertilization, grapes can have three different seedless parthenocarpy types…”.

-          Improve cohesion between the results presented and the conclusions drawn in the discussion section.

Author Response

Response to Reviewer 1 comments

First of all, we really appreciated the valuable comments for our manuscript (IJMS-2929742) from reviewers and editors. We carefully and completely revised our manuscript based on the valuable comments from you and Reviewers, and hope the modified manuscript could meet the requirements of publishing in IJMS. All the responses to the comments were showed in the following pages, and the revised contents in our manuscript were in red color in the updated text. The article is modified in revision mode. All the authors have approved the revised manuscript. Here are out point-to-point response letter.

Point 1: Identify and correct any missing commas, periods, and italics throughout the document. 

Response 1: Thanks for your careful suggestion. Following your advices, we corrected commas, periods, and italics in the article throughout the whole manuscript.

Point 2: Review references older than 5-10 years and update them with more recent sources. Eliminate references from older years (1994, 2000, 2004, 2008, 2003, 2002, 2006, 1995, 1997, 1998) to reflect current rese.

Response 2: Thanks for your good suggestion. We carefully re-examined all the old references and updated them to use the latest information as suggested.

Point 3: Significantly enhance figure resolution to improve readability. Address inconsistencies in figure formatting, such as font types, sizes, and background colors, to ensure visual cohesion across all figures.

Response 3: Thanks for your constructive suggestion. According to your suggestion, we improved chart resolution, adjusted font type, size and background color to ensure visual consistency across all charts.

Point 4: The discussion section needs a clear and concise overview of the topic. I suggest considering the rephrasing of sentences to make them more concise and focused, maintaining direct links between the discussion and the results. Furthermore, reorganizing the information in a manner that creates a logical flow.

Response 4: Thanks for your suggestion. Following your advices, we have reworked the discussion section to make the language more concise and focused, to strengthen the link between the discussion and the results, and to make the information presented more logical.

Point 5: Enhance the conclusion by providing a comprehensive summary of findings and their implications. Elucidate tangible implications of the study's results beyond a mere recapitulation of the article's content. I would suggest to expand the discussion on the practical implications of the findings for the wine industry and agriculture in general.

Response 5: Thanks for your constructive suggestions. Following your advices, we have comprehensively summarized the results of the study, clarified the related research implications, and strengthened the conclusions. Moreover, we supplemented the practical implication of our findings for the wine industry and agriculture in general, as suggesed.

Point 6: Shorten the abstract summarizing the key points of the article without unnecessary verbosity

Response 6: Thanks for your constructive suggestion. As you suggested, we carefully shortened the abstract to succinctly summarize the main points of the article and highlight the topic.

Point 7: Provide references for statements made in the paragraph of the introduction section“Based on the conditions of pollination and fertilization, grapes can have three different seedless parthenocarpy types...”)

Response 7: Thanks for your constructive suggestion. Following your advices, we provided the related references for the introductory section.

Point 8: Improve cohesion between the results presented and the conclusions drawn in thediscussion section.

Response 8: Thanks for your constructive suggestion. According to your suggestions, we carefully revised and re-phrassed the related contents to strengthen the coherence between the results of the discussion section and the conclusions reached.

Reviewer 2 Report

Comments and Suggestions for Authors

Seedlessness is a lucrative trait, especially in grape. It is well established that MADS-box gene Agamous-like genes involve in seed development (In grape there are several articles on MADS-box gene involvement in seed formation). In this prospective deep study on AGLs is significant. I was interested to read the MS. Unfortunately, it was hard for me to understand both in terms of presentation and language. Moreover, the information was presented in DISCUSSION and Conclusion is largely known.  Every section of the MS needs to be improved. Introduction did not build a strong argument for the importance of this research. Results sounds like a discussion. Methodology is incomplete, and it is difficult to correlate with the results.  There is nothing in conclusion except reputation of the known information.  References are not correctly cited.

Comments on the Quality of English Language

Must be improved

Author Response

Response to Reviewer 2 comments

First of all, we really appreciated the valuable comments for our manuscript (IJMS-2929742) from reviewers and editors. We carefully and completely revised our manuscript based on the valuable comments from you and Reviewers, and hope the modified manuscript could meet the requirements of publishing in IJMS. All the responses to the comments were showed in the following pages, and the revised contents in our manuscript were in red color in the updated text. The article is modified in revision mode. All the authors have approved the revised manuscript. Here are out point-to-point response letter.

Point:Seedlessness is a lucrative trait, especially in grape. It is well established that MADS-box gene Agamous-like genes involve in seed development (In grape there are several articles on MADS-box gene involvement in seed formation). In this prospective deep study on AGLs is significant. I was interested to read the MS. Unfortunately, it was hard for me to understand both in terms of presentation and language. Moreover, the information was presented in DISCUSSION and Conclusion is largely known.  Every section of the MS needs to be improved. Introduction did not build a strong argument for the importance of this research. Results sounds like a discussion. Methodology is incomplete, and it is difficult to correlate with the results.  There is nothing in conclusion except reputation of the known information.  References are not correctly cited.

Response:Thank you for your suggestion. Based on your advices, we have reread the content of the article and sorted out the presentation of the content, made the language more concise and focused, and reworked the content of the conclusion section, improved the methodology to enhance the logic between the introduction and conclusion, expanded the scope of the discussion section, and corrected the citation of references.
